# Upgrading the Fusion of Imprecise Classifiers

**DOI:** 10.3390/e25071088

**Published:** 2023-07-19

**Authors:** Serafín Moral-García, María D. Benítez, Joaquín Abellán

**Affiliations:** Department of Computer Science and Artificial Intelligence, University of Granada, 18012 Granada, Spain

**Keywords:** imprecise classification, Credal Decision Trees, ensembles, bagging, combination technique

## Abstract

Imprecise classification is a relatively new task within Machine Learning. The difference with standard classification is that not only is one state of the variable under study determined, a set of states that do not have enough information against them and cannot be ruled out is determined as well. For imprecise classification, a mode called an Imprecise Credal Decision Tree (ICDT) that uses imprecise probabilities and maximum of entropy as the information measure has been presented. A difficult and interesting task is to show how to combine this type of imprecise classifiers. A procedure based on the minimum level of dominance has been presented; though it represents a very strong method of combining, it has the drawback of an important risk of possible erroneous prediction. In this research, we use the second-best theory to argue that the aforementioned type of combination can be improved through a new procedure built by relaxing the constraints. The new procedure is compared with the original one in an experimental study on a large set of datasets, and shows improvement.

## 1. Introduction

The standard classification task tries to predict one state of a class variable when a new item or instance appears. A classifier normally learns from a set of data. In situations where there is not enough information to point out a unique state of the class variable, Imprecise classification arises. In this case, a set of states of that variable are predicted, which are the states that have no clear information against them. It can be said that Imprecise Classification discards some states and considers the ones that cannot be discarded via the available information.

Credal Decision Tree (CDT) [1,2] is a single decision tree that uses imprecise probability models and uncertainty-based information measures in the tree-building process, specifically, the maximum of entropy measure. This classifier has been used, similar to other comparable models, in many ensemble procedures [3,4], and has been adapted for Imprecise Classification (ICDT) [5]. However, ICDT was not the first imprecise classifier presented in the literature; the *Naive Credal Classifier* (NCC) [6] was the first imprecise classifier presented. It combines the naive assumption (i.e., all the attributes are independent given the class variable) and the Imprecise Dirichlet Model (IDM) to produce an imprecise classification. This NCC procedure provides worse results than the ICDT model [5], and we do not use it here.

An imprecise classifier might obtain a set of states of the class variable, often called the set of *non-dominated states*. It is constituted by those class states for which there is no other “better” choice according to an established criterion, usually known as the *dominance criterion*. Intuitively, an Imprecise Classification evaluation measure must consider whether the real class value belongs to the non-dominated states set as well as the precision of the predicted set of class values, which is measured based on its cardinality.

Imprecise predictions tend to consist of sets of states. Therefore, it is not trivial to combine multiple imprecise predictions. Until recently there was no technique for this. If the predictions are not properly combined, then it is highly likely that the ensemble will perform no better than a unique classifier, as excessive information reduction may result.

The first ensemble of imprecise classifiers was recently presented by Moral-García et al. [7]. This ensemble is based on a bagging scheme [8], which has obtained satisfactory performance in precise classification, especially when used with CDTs. For these reasons a bagging scheme with ICDTs was introduced in [7]. The proposed technique for combining imprecise predictions aims to maximize the precision of the bagging scheme. If the combination procedure is too conservative, then little information will be obtained, and the performance of a single classifier might be worsened. The procedure of [7] only considers the class values with the minimum level of dominance. Hence, it could be considered that such a procedure assumes too much risk, although it achieves good results when compared with the simple ICDT model.

In a procedure for combining imprecise classifiers, we consider the set of states belonging to the set of the minimum level of dominance as the winner states. In the optimal case of a clear set of winners states, i.e., if the difference with respect to the others is relatively large, the minimum level of dominance could be considered a very good procedure. The problem is that in the majority of real cases this does not occur. Consequently, the risk that we must assume can produce bad informative results. This information is obtained via the measure that we need to apply to quantify the performance.

Here, we consider the trade-off between risk and success in the procedure proposed in [7] to be less than ideal. Thus, our starting point is to reduce the risk that we must assume in a procedure to combine imprecise classifiers. This can be considered as a reduction of the constraints taken into account to improve the level of information that such a type of procedure should offer. When the optimal conditions cannot be satisfied, we can apply the *theory of the second-best* [9], which was originally presented in the area of economics.

Considering the reasons above exposed, we present a new procedure for combining imprecise classifiers that reduces the constraint of the minimum level of dominance, i.e., we consider a scenario in which the optimal situation is not normally attained. In the new procedure, we add a parameter to control the degree of risk we are willing to take. If we decrease that degree, the risk is higher, while the information can be higher as well. Hence, we find it interesting to consider a trade-off between risk and information. This is obtained by relaxing the minimum level of dominance considered by the original method through a parameter. Greater values of that parameter provide less information; however, they provide less risk as well, because a larger set of class values can be obtained.

The new procedure presented here can provide a greater amount of information via the measure for this aim presented in [5]. In this paper, exhaustive experimentation on 34 different datasets is carried out. All these datasets have at least three states of the class variables to ensure that our research on Imprecise Classification makes more sense. We compare the procedure presented in [7] with our new proposal. To compare the results, we apply the measure from [5]. This experimentation shows that the new procedure, which is more cautious in its predictions, obtains clearly better results based on this measure. To reinforce this assertion, our statistical tests show a performance improvement in favor of the new procedure with a strong level of significance.

The rest of this paper is organized as follows: in Section 2, the Imprecise Credal Decision Tree algorithm and the existing bagging scheme for Imprecise Classification are described; Section 2.4 introduces our proposed procedure of combining imprecise classifiers; and the experimental study carried out in this work is detailed in Section 4. Finally, Section 5 provides conclusions and ideas for future work.

## 2. Background

Let *C* be the class variable and let $\left\{c_{1},c_{2},\cdots ,c_{K}\right\}$ be its set of possible values.

### 2.1. Imprecise Credal Decision Tree

The Imprecise Credal Decision Tree (ICDT) algorithm introduced in [5] adapts the Credal Decision Tree method for Imprecise Classification. The tree-building process of both methods is identical.

Let D be the dataset corresponding to a certain node. Suppose that such a dataset contain $N^{D}$ instances. Let $n^{D}\left(c_{j}\right)$ be the number of instances in D that satisfy $C=c_{j},$∀j=1,2,⋯,K. In order to represent the uncertainty-based information about *C* in D, ICDT uses the *Imprecise Dirichlet Model* (IDM) [10], a formal imprecise probability model based on probability intervals. Specifically, the IDM predicts that the probability that *C* takes each possible value $c_{j}$ belongs to the following interval:(1)$$I^{D}\left(c_{j}\right)=\left[\frac{n^{D}\left(c_{j}\right)}{N^{D}+s},\frac{n^{D}\left(c_{j}\right)+s}{N^{D}+s}\right],\;\forall j=1,2,\cdots ,K,$$
where s>0 is a given parameter of the model that indicates the estimated degree of imprecision in the data. This set of intervals yields the following credal set (a credal set being a convex and closed set of probability distributions) on *C* [11]:(2)$$P^{D}\left(C\right)=\left\{p\in P\left(C\right)|p\left(c_{j}\right)\in I^{D}\left(c_{j}\right),\;\forall j=1,2,\cdots ,K\right\},$$
where P(C) is the set of all probability distributions on *C*. ICDT quantifies the uncertainty about *C* through the maximum entropy on $P^{D}\left(C\right)$:(3)$$S^{*}\left(P^{D}\left(C\right)\right)=\max_{p\in P^{D}\left(C\right)}S\left(p\right),$$
where S(p) is the Shannon entropy [12] of the probability distribution *p*, determined by
(4)$$S\left(p\right)=-\sum_{j=1}^{K}p\left(c_{j}\right)\log_{2}p\left(c_{j}\right).$$

It must be remarked that the maximum entropy is a well-established uncertainty measure on credal sets, and satisfies the essential mathematical properties [13].

Let *X* be an attribute taking the possible values $\left\{x_{1},\ldots ,x_{t}\right\}$. The split criterion of ICDT, called the *Imprecise Information Gain* (IIG), is defined as follows:(5)$$IIG^{D}\left(C\right)=S^{*}\left(P^{D}\left(C\right)\right)-\sum_{i=1}^{t}p^{D}\left(x_{j}\right)S^{*}\left(P^{D}\left(C|X=x_{i}\right)\right),$$
where $p^{D}\left(x_{j}\right)$ is the probability of $X=x_{j}$ according to the probability distribution that leads to the maximum entropy on the IDM credal set on *X* on D, while $P^{D}\left(C|X=x_{i}\right)$ is the IDM credal set on *C* on the subset of D constituted by those instances such that $X=x_{i},\forall i=1,2,\ldots ,t$.

ICDT differs from CDT in the criterion used to make a prediction at a leaf node. While CDT uses majority vote, ICDT applies a dominance criterion to the IDM probability intervals to obtain the set of non-dominated states at a leaf node.

Let L be a leaf node; let $N^{L}$ be the total number of instances in L and let $n^{L}\left(c_{j}\right)$ be the number of instances in L for which $C=c_{j},\forall j=1,2,\ldots ,K$. Then, we have the following set of IDM probability intervals at L:(6)$$\left\{\left[\frac{n^{L}\left(c_{j}\right)}{N^{L}+s},\frac{n^{L}\left(c_{j}\right)+s}{N^{L}+s}\right],\;j=1,2,\cdots ,K\right\}.$$

ICDT uses the stochastic dominance criterion on this set of intervals, which asserts that a class value $c_{j}$ dominates another one $c_{k}$ if and only if
$$\begin{array}{c} \frac{n^{L}\left(c_{j}\right)}{N^{L}+s}\geq \frac{n^{L}\left(c_{k}\right)+s}{N^{L}+s}\Leftrightarrow n^{L}\left(c_{j}\right)\geq n^{L}\left(c_{k}\right)+s. \end{array}$$

According to the results proved in [14], stochastic dominance is the well-established dominance criterion for IDM probability intervals.

### 2.2. Bagging of Credal Decision Trees for Imprecise Classification

Thus far, the only combined method for imprecise classification is called the Bagging of Imprecise Credal Decision Trees (Bagging-ICDT) method [7]. The idea of this algorithm is similar to the bagging scheme for standard classification. For each base classifier, a bootstrapped sample of the training set is first chosen; then, using the selected sample and ICDT, an Imprecise Classification model is learned.

The key issue with Bagging-ICDT is combining the predictions of the base imprecise classifiers. We remark here that this is not a trivial question, as imprecise classifiers predict a set of non-dominated states; in fact, there are multiple ways of combining multiple imprecise predictions, as certain classifiers may predict that a class value will be dominated while others predict it will be non-dominated. The fundamental point is to find a trade-off between *information* and *risk*, where the term *information* indicates the precision of the prediction, i.e, the number of non-dominated states set, and the term *risk* refers to the probability that the real class value does not belong to the set of non-dominated states. Obviously, a high level of information gives rise to a high level of risk.

The combination technique proposed in [7] tries to ensure that the ensemble scheme is as informative as possible, though it implies a higher risk of misclassification. In this approach, to classify an instance, the number of classifiers that predict such a state as dominated is counted for each class state. The set of non-dominated states predicted by Bagging-ICDT is composed of those class values predicted as dominated by the minimum number of classifiers.

Figure 1 summarizes the Bagging-ICDT method proposed in [7].

The problem with this procedure is that it is possible that the set of states with the minimum level of dominance has support very close to one of the other states. In this case, the level of risk is very high. This situation appears in the majority of real cases. Hence, the risk that we must assume can produce bad informative results. We find it interesting to consider a trade-off between risk and information by relaxing the constraint represented by the minimum level of dominance.

### 2.3. Evaluation Metrics for Imprecise Classification

To evaluate the performance of imprecise classifiers, the MIC evaluation measure [5] can be employed. This metric takes into account the possibility that in certain situations the errors may have different degrees of importance. Here, consider the same level of importance for all errors, as the different weights of each error must generally be quantified by experts in a particular area.

MIC is defined as follows:(7)$$MIC=\frac{1}{N_{tst}}\left(\sum_{i:Success}\log\frac{\left|U_{i}\right|}{K}+\frac{1}{K-1}\sum_{i:Error}\log K\right),$$
where $N_{tst}$ is the number of test instances, *K* is the number of class states, and $\left|U_{i}\right|$ is the predicted set of non-dominated states for the *i*-th test instance, ∀i=1,2,…,N_tst.

It is obvious that the higher the value of MIC, the better the performance. We can observe that the optimal value of MIC, which is reached when all the predictions are precise and correct, is equal to $-\log\frac{1}{k}=\log k$. Moreover, when it is verified that $\left|U_{i}\right|=k$, $\forall i=1,\cdots ,N_{Test}$, i.e, when the imprecise classifier always predicts all possible class values as non-dominated states for a given instance, the value of MIC is equal to 0. This is intuitively more correct, as in these cases the classifier is not informative.

We find that this measure is the better one to use, as it does not have the problems presented by others, as can be seen in [7].

### 2.4. The Second-Best Theory

The theory of the second-best [9] was first presented in the area of economics, and essentially stands as follows:

“It concerns the situation when one or more optimality conditions cannot be satisfied: if one optimality condition in an economic model cannot be satisfied, then the next-best solution might involve changing other variables away from the values that would otherwise be optimal. Politically, the theory implies that if it is infeasible to remove a particular market distortion, introducing one or more additional market distortions in an interdependent market may partially counteract the first, and lead to a more efficient outcome.”

When a market distortion appears or is introduced (i.e., optimal conditions do not exist), then the application of the best approach for those conditions, known as the second-best (i.e., not optimal) can produce clear advantages. This is known as the *second-best equilibrium*.

## 3. Application of the Second-Best Theory to the Combination of Imprecise Classifiers

In our case, the trade-off between risk and accuracy when combining the results of different imprecise classifiers can produce a similar situation. As said above, the difference between the set of winners and the rest of the states may not be large enough to consider that set of winners to be a good result. We have found that this excessive risk can produce a decrease in accuracy and a clear decrease in information via the MIC measure, which takes into account the errors in the results in an important way. Hence, we think that it is necessary to limit the risk in favor of accuracy. This is where we consider the second-best set of non-dominated states.

Based on the above assessments, we consider the need to add a constraint for rejecting states with similar information in the set of non-dominated states during the combination process. We think that at least those states very close to the one selected in the original method must be taken into account. We illustrate this issue via the following example.

**Example 1.** 
*Consider a case where the class variable has five states $\left\{c_{1},c_{2},\cdots ,c_{5}\right\}$ and a bagging procedure is applied using the standard size of 100 trees. To simplify, we can suppose that $c_{1}$ has the minimum number of votes against; in this case that value is 50 (dominated for 50 trees). However, $c_{2}$ has 52 votes against, while $c_{3}$, $c_{4}$, and $c_{5}$ each have 80 votes against. The original method of combining the minimum dominated states outputs the set $\left\{c_{1}\right\}$, which represents excessive risk when taking the real situation into account.*

*In the resulting situation, the set we consider the optimal result is clearly $\left\{c_{1},c_{2}\right\}$ because of its small difference in votes against with respect to the set $\left\{c_{1}\right\}$.*

*We can analyze possible situations associated with the above numbers via the MIC measure. We consider extreme situations around the number of votes against in concordance with those numbers (the example numbers cannot be the most extreme ones, but they are very close to the extreme possible values) in a bagging procedure of 100 trees to calculate the maximum and minimum possible degrees of information obtained via the MIC measure.*

1.
*The result is $\left\{c_{1}\right\}$: the best situation (B1) is 50 success, i.e., 50 trees where $c_{1}$ is the only state non-dominated and 50 errors with that result, i.e., the set of non-dominated values is in the set $\left\{c_{2},\cdots ,c_{5}\right\}$. By this, we mean that with these values the possible real state is always accurate when the resulting set is $\left\{c_{1}\right\}$, i.e., no other option can be reached. For the cases where the result is, for example, the set $\left\{c_{1},c_{3}\right\}$, the real state associated with the instance can be $c_{3}$. The worst situation (W1) is two trees, where $c_{1}$ is the only non-dominated state (because those where $c_{2}$ is non-dominated could coincide with the 48 where $c_{1}$ is non-dominated as well), and 98 errors, i.e., the set of non-dominated values is in the set $\left\{c_{2},\cdots ,c_{5}\right\}$.*
2.
*The result is $\left\{c_{1},c_{2}\right\}$: the best situation (B2) is 98 success, i.e., 50 trees where $c_{1}$ and $c_{2}$ are in the set of non-dominated states but never coincide, and two errors, i.e., the set of non-dominated values of those two trees is in the set $\left\{c_{3},c_{4},c_{5}\right\}$. The worst situation (W2) is 50 trees where $c_{1}$ is one of non-dominated states and $c_{2}$ is non-dominated by the same trees, and 50 errors, i.e., the set of non-dominated values of those 50 trees is in the set $\left\{c_{3},c_{4},c_{5}\right\}$.*


*The idea here is to think about the possible real value of the class variable associated with an instance while considering the numbers discussed above. Using the MIC measure, we can obtain the following values:*

*Using MIC measure (Equation 7):*

(8)
$$MIC=\frac{1}{N_{tst}}\left(\sum_{i:Success}\log\frac{\left|U_{i}\right|}{K}+\frac{1}{K-1}\sum_{i:Error}\log K\right)$$


*The values compatible with the above situations are*



$MIC\left(A1\right)=\frac{1}{100}\left(-50\log\left(\frac{1}{5}\right)+\frac{50}{4}\log\left(5\right)\right)=1,006$



$MIC\left(W1\right)=\frac{1}{100}\left(-2\log\left(\frac{1}{5}\right)+\frac{98}{4}\log\left(5\right)\right)=0,413$



$MIC\left(A2\right)=\frac{1}{100}\left(-98\log\left(\frac{2}{5}\right)+\frac{2}{4}\log\left(5\right)\right)=0,906$


*$MIC\left(W2\right)=\frac{1}{100}\left(-50\log\left(\frac{2}{5}\right)+\frac{50}{4}\log\left(5\right)\right)=0,659$.*


*We can observe that from one result to the other the upper value number is reduced by around 10% (from 1,006 to 0,906), whereas the lower value is increased by around 60% (from 0,413 to 0,659). In terms of bets, the upper value could be considered as the maximum win (greater values imply the possibility of a greater win) and the lower one as complement of the maximum loss or maximum risk (greater values imply lower levels of risk).*

*In this case, for a decrease in the possibility of the maximum win of 10%, the risk can be reduced by around 60%. Hence, in our opinion, the more conservative strategy of considering the set $\left\{c_{1},c_{2}\right\}$ in stead of $\left\{c_{1}\right\}$ makes more sense for predicting the real value of the class variable.*


In Example 1, it can be seen that the use of the second-best theory can help us to obtain an increase of information expressed in form of extreme values of gains. We can consider that we are introducing the constraint of not taking on excessive risk, even this results in a reduced maximum probability of success. Clearly, in this example, considering the second-best issue $\left\{c_{1},c_{2}\right\}$ instead of $\left\{c_{1}\right\}$ notably reduces the likelihood of loss.

For our combination procedure, the problem is now how to quantify the optimal difference among the best states to determine the maximum difference of votes against in terms of percentage. If we call this percentage γ%, then the method explained in Section 2.2 is modified only in terms of the procedure provided in Figure 2.

Here, the parameter γ represents the degree of risk we are willing to take. If we decrease its value, the risk is higher, though the information can be higher as well. We consider it important to use a trade-off between risk and information. Greater values of γ provide less information with less risk because a larger set of class values can be obtained.

## 4. Experimentation

### 4.1. Experimental Settings

For experimentation, we used the implementation in the Weka software [15] (https://en.wikipedia.org/wiki/Weka_(software)) for ICDT, as used in [5]. The structures provided in Weka for the bagging scheme were utilized to add all the necessary methods for implementing the bagging of our ICDT procedures. We call the method presented in [7] *Bagging-ICDT* and the new method presented here with a parameter γ=5% *Bagging-ICDT2B*, which can be considered a good trade-off between information and risk. The bagging schemes were all applied with 100 trees, a number of classifiers established as suitable for bagging [16].

Both methods were tested on 34 known datasets found in the *UCI Machine Learning repository* [17]. These datasets are varied with regard to the number of instances, number of discrete and continuous features, ranges of values of discrete attributes, number of states of the class variable, etc. Consistent with the experimental analysis carried out in [5], in all selected datasets the class variable had at least three possible values, as with only two class values all of them are either predicted or a single one. Table 1 shows the most relevant characteristics of each dataset.

In accordance with the experiments carried out in [7], missing values were replaced by modal (mean) values for discrete (continuous) features. Then, continuous attributes were discretized by means of Fayyad and Irani’s discretization procedure [18]. A ten-fold cross-validation procedure was repeated ten times for each preprocessed dataset.

For statistical comparisons, following the recommendations in [19] when there are two algorithms, the Wilcoxon test [20] was used with a level of significance of α=0.01. For each dataset, this test ranks the differences between the performance of two methods regardless of the signs. Then, the ranks for the positive differences are compared with those for negative differences.

### 4.2. Results

Table 2 shows the MIC results obtained by each algorithm on each dataset. A summary of the MIC results is presented in Table 3, which shows the average value, the result of the Wilcoxon test, and the number of datasets where one algorithm beats the other algorithm as exspressed by the MIC measure.

Considering the MIC results in Table 2, Bagging-ICDT obtains lower values than Bagging-ICDT2B. While the difference is not extremely large, the values are almost always in favor of the new procedure. The differences are greater on datasets with a large number of states of the class variable, which makes sense due to the range of the MIC measure. The number of wins for each method compared to the other one can be seen in the third row of Table 3. There is a large difference in the wins of the new procedure compared to the original one (28 vs. 3).

The most important comparison can be seen in the second row of Table 3, where the results of the Wilcoxon test are presented. This test was carried out with a very strong level of significance of 0.01. Here, we can observe that Bagging-ICDT2B significantly outperforms the original Bagging-ICDT method, demonstrating that the trade-off between risk and information in the original method is not optimal.

## 5. Conclusions and Future Work

In this work, we have shown that the first method to combine imprecise classifiers based on the minimum level of non-dominance, while a very strong method of combination, has the important drawback of possible information loss. This is because the difference between the set of winning states and the rest can be very small in real situations. Here, we have used the second-best theory to argue that this method can be improved to obtain better results in the combination of imprecise classifiers by relaxing the constraints taken into account.

The experimental analysis carried out in our research has revealed that the new method presented here is clearly superior to the original method. The new method outperforms the existing one on a statistical test with a high level of significance. It has been demonstrated that better results can be produced by relaxing the constraints taken into account in the original method to minimize the loss of information at the cost of assuming greater risk, as the trade-off between risk and information in the original method is not optimal.

The new method presented here can be tuned for specific datasets to obtain better results in future work. Moreover, the procedure presented here for imprecise classification can be inserted in other known ensemble schemes, such as Boosting [21] or Random Forest [8].

## Figures and Tables

**Figure 1 entropy-25-01088-f001:**
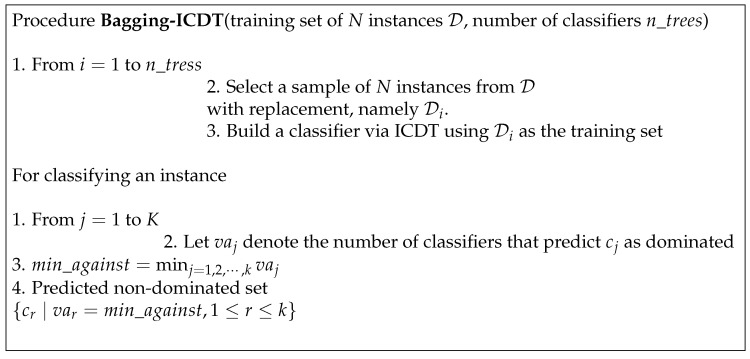
Summary of the Bagging-ICDT method.

**Figure 2 entropy-25-01088-f002:**
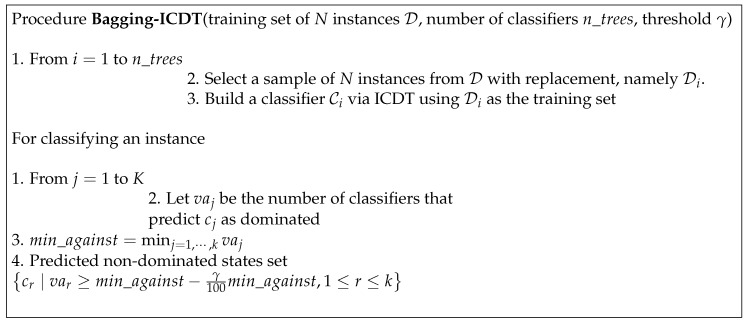
New bagging scheme with ICDT: Bagging-ICDT2B.

**Table 1 entropy-25-01088-t001:** Dataset description. Column “N” is the number of instances, column “Feat” is the number of features or attribute variables, column “Cont” is the number of continuous variables, column “Disc” is the number of discrete variables, column “K” is the number of states of the class variable, and column “Range” is the range of value of the discrete attributes.

Dataset	N	Feat	Cont	Disc	K	Range
anneal	898	38	6	32	6	2–10
arrhythmia	452	279	206	73	16	2
audiology	226	69	0	69	24	2–6
autos	205	25	15	10	7	2–22
balance-scale	625	4	4	0	3	-
bridges-version1	107	11	3	8	6	2–54
bridges-version2	107	11	0	11	6	2–54
car	1728	6	0	6	4	3–4
cmc	1473	9	2	7	3	2–4
dermatology	366	34	1	33	6	2–4
ecoli	366	7	7	0	7	-
flags	194	30	2	28	8	2–13
hypothyroid	3772	30	7	23	4	2–4
iris	150	4	4	0	3	-
letter	20,000	16	16	0	26	-
lymphography	146	18	3	15	4	2–8
mfeat-pixel	2000	240	0	240	10	4–6
nursery	12,960	8	0	8	4	2–4
optdigits	5620	64	64	0	10	-
page-blocks	5473	10	10	0	5	-
pendigits	10,992	16	16	0	10	-
postop-patient-data	90	9	0	9	3	2–4
primary-tumor	339	17	0	17	21	2–3
segment	2310	19	16	0	7	-
soybean	683	35	0	35	19	2–7
spectrometer	531	101	100	1	48	4
splice	3190	60	0	60	3	4–6
sponge	76	44	0	44	3	2–9
tae	151	5	3	2	3	2
vehicle	946	18	18	0	4	-
vowel	990	11	10	1	11	2
waveform	5000	40	40	0	3	-
wine	178	13	13	0	3	-
zoo	101	16	1	16	7	2

**Table 2 entropy-25-01088-t002:** Complete results obtained for the MIC measure, with the best results marked in bold.

Dataset	Bagging-ICDT	Bagging-ICDT2B
anneal	1.7847	1.7847
arrhythmia	1.9316	**1.9440**
audiology	2.5936	**2.6132**
autos	1.5553	**1.5618**
balance-scale	0.6006	**0.6049**
bridges-version1	**1.0446**	1.0340
bridges-version2	0.9767	**0.9910**
car	1.2568	**1.2576**
cmc	0.2636	**0.2682**
dermatology	**1.6844**	1.6834
ecoli	1.6182	**1.6208**
flags	1.1398	**1.1615**
hypothyroid	1.3744	**1.3746**
iris	0.9982	**0.9985**
letter	2.6771	**2.6967**
lymphography	0.9417	**0.9558**
mfeat-pixel	2.0066	**2.0208**
nursery	1.5398	**1.5405**
optdigits	1.9579	**1.9715**
page-blocks	1.5418	**1.5424**
pendigits	2.0925	**2.0966**
postoperative-patient-dat	0.6207	0.6207
primary-tumor	1.2278	**1.2504**
segment	1.8331	**1.8344**
soybean	2.7203	**2.7256**
spectrometer	1.9527	**2.0219**
splice	1.0077	**1.0080**
sponge	1.0121	**1.0133**
tae	0.2216	0.2216
vehicle	0.8372	**0.8417**
vowel	1.8594	**1.8680**
waveform	0.7325	**0.7354**
wine	0.9817	**0.9837**
zoo	**1.8578**	1.8571
Average	1.4248	**1.4325**

**Table 3 entropy-25-01088-t003:** Summary of the results obtained for the MIC measure. In the “Wilcoxon test” rows, if one classifier is significantly better than the other one this is expressed by “*”. The level of significance used is 0.01. The row “Number of wins” indicates the number of datasets on which one algorithm beat the other in terms of MIC.

		Bagging-ICDT	Bagging-ICDT2B
MIC:	Average	1.3652	1.4248
	Wilcoxon *t*-test		*
	Number of wins	3	28

## Data Availability

Not applicable.

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
