# Peer review of "Upgrading the Fusion of Imprecise Classifiers"

_entropy, 2023, doi:10.3390/e25071088_

Round 1
Reviewer 1 Report
This paper proposes a program for imprecise classifiers based on the combination of least dominant levels. The second-best theory improves the combination type, and the constraints are relaxed, to obtain the best results of the imprecise classifier combination. The idea is interesting and useful. However, the quality of the paper needs to be improved. The specific comments are as follows:
1. On page 8, is the threshold γ a subjective parameter? If it is subjective, I suggest the authors give some explanation.
2. The method proposed in this article involves the second theory, and I suggest that the authors supplement the relevant knowledge in the second part.
3. In the actual data set, how do the supports and disadvantages of different classifiers get it? I suggest that the authors add some details of the experiment.
4. I suggest that the authors make more comparisons with the work related to imprecise classification in recent years. What is the great value and meaning than the other methods? Why it could give a more considerable result?
5. I recommend that authors refer to more recent literature.
6. I suggest that the authors add some graphics to enhance the readability of the article.
7.On page 6, the authors should further prove why the value of the parameter γ is 5%.
8. In addition, there are some details to pay attention to. As follows:
(1)On page 2, the authors should indicate who proposed the first ensemble of imprecise classifiers in order to make the paper more rigorous.
(2)On page 5, the authors should make Example 1 clearer, and it would be better if it could be combined with the formula.
(3)On page 6, the authors should have the same format for these equations to make the paper more rigorous.
(4)The authors should be careful to check this paper, such as lines 175 on page 6.
Based on the above comments, I think the content of the paper still needs to be improved, and I will make a major revision.
This manuscript is readable but still requires careful grammar and details grooming.
Reviewer 2 Report
In general this paper highlights an actual topic: Credal sets (decision trees)
The paper --is based on the content -- quite interesting and triggers an actual reserach topic which is discussed in many paper. Uncertainty in (weak) classifiers. This topic is in the focus of the research community whereas aleatoric and epistemic uncertainty is well known research.
Insofar the content is significant.
However, the paper must be revised because of inappropriate self-citations of the corresponding author. Indeed, 10 of 22 citations are self-citations.
Furthermore, all argumenations pro a new concept are based on self-citations, e.g. line 61,62, and 65; 87, 108 and so forth. Argumations and proofs are mainly based on self-citations.
Furthermore, some citations seems to be missed:
E. Hüllermeier and W. Waegeman. Aleatoric and epistemic uncertainty in machine learning: An introduction to concepts and methods. Machine Learning, 110(3):457–506, 2021. doi: 10.1007/s10994-021-05946-3.
V.-L. Nguyen, M. H. Shaker, and E. Hüllermeier. How to measure uncertainty in uncertainty sampling for active learning. Machine Learning, pages 1–34, 2021.
R. Yager. Entropy and specificity in a mathematical theory of evidence. International Journal of General Systems, 9: 249–260, 1983.
Bernard, J. M. (2005). An introduction to the imprecise Dirichlet model for multinomial data. International Journal of Approximate Reasoning, 39(2–3), 123–150.
De Campos, L. M., Huete, J. F., & Moral, S. (1994). Probability intervals: A tool for uncertain reasoning. International Journal of Uncertainty, Fuzziness and Knowledge-Based Systems, 2(02), 167–196.
The paper should be revised in general to get rid of the self-citations. Moreover, the underlying proofs must be shown more clearly without self-citations.
The English language is in order (minor typos).
Reviewer 3 Report
In this manuscript, the authors adopted the second-best theory to combine the predictions generated by multiple imprecise classifiers. The authors should address the following issues.
1. The Abstract should be reorganized. The authors should avoid introducing fundamental knowledge, such as Lines 4-11. The authors should briefly introduce the research background and identify the research objective, followed by the research method. Most importantly, the research result is totally unclear. What did the authors mean ‘experimental results show that assertion’?
2. Keywords are missing.
3. In Section 1, the authors failed to review SOTA work about the fusion of imprecise predictions. What is the research gap?
4. The main contributions of this manuscript are unclear. It is suggested that the authors should list bullet points to state their newly-introduced knowledge at the end of the Introduction Section.
5. In the Background Section, there lacks a problem statement. It is not reasonable to directly state the methods without providing a problem to be solved.
6. Did Example 1 have any practical meaning? Because defining the authors’ own case can hardly prove the effectiveness of the authors’ proposal. If this example has been used in any literature or open-access dataset, the references should be cited.
7. The theoretical details of Section 3 should be supplemented. How did the authors take advantage of the methods presented in Section 2?
8. The experimental design is too simple. The authors did not compare their Bagging-ICDT2B with any other previous work. The effectiveness of ICDT2B cannot be proved.
9. The authors should carefully check their cited references. The latest one is from 2014, which is totally unacceptable. The authors should check for recent publications (e.g., within 5 years).
Extensive English editing is needed.
Round 2
Reviewer 1 Report
All my comments have been covered, and I recommend accepting the paper.
Reviewer 2 Report
The form now available has been significantly improved in the text section and is thus easier to read. The intention of the authors is comprehensible; the procedure as such is also comprehensible.
The experimental part gives a first insight, but a more intensive experimental consideration would certainly have helped but this is no reason to reject the paper.
However, the high number of self-citations of the corroborating author remains fundamentally problematic. It is understandable that the cited sources are given to justify the procedure and to present the approach. However, it remains that further supporting sources, e.g. on second-best equilibrium, are necessary to ensure that the number of self-citations does not exceed 1/3 of all citations.
Of course, it is understandable and acknowledged by the reviewer that much original work is by the corresponding author, but the present work is an extension of that which has gone before, so that special attention must be paid to the sources here.
Reviewer 3 Report
I have no further comments.
Moderate editing of English language required